# SLC35F2, a Transporter Sporadically Mutated in the Untranslated Region, Promotes Growth, Migration, and Invasion of Bladder Cancer Cells

**DOI:** 10.3390/cells10010080

**Published:** 2021-01-06

**Authors:** Roland Kotolloshi, Martin Hölzer, Mieczyslaw Gajda, Marc-Oliver Grimm, Daniel Steinbach

**Affiliations:** 1Department of Urology, Jena University Hospital, 07740 Jena, Germany; roland.kotolloshi@med.uni-jena.de (R.K.); marc-oliver.grimm@med.uni-jena.de (M.-O.G.); 2RNA Bioinformatics and High-Throughput Analysis, Friedrich Schiller University Jena, 07743 Jena, Germany; martin.hoelzer@uni-jena.de; 3Department of Forensic Medicine, Section of Pathology, Jena University Hospital, 07740 Jena, Germany; mieczyslaw.gajda@med.uni-jena.de

**Keywords:** bladder cancer, oncogene, SLC35F2, UTR mutation, progression

## Abstract

Bladder cancer is a very heterogeneous disease and the molecular mechanisms of carcinogenesis and progression are insufficiently investigated. From the DNA sequencing analysis of matched non-muscle-invasive bladder cancer (NMIBC) and muscle-invasive bladder cancer (MIBC) samples from eight patients, we identified the tumour-associated gene SLC35F2 to be mutated in the 5′ and 3′ untranslated region (UTR). One mutation in 3′UTR increased the luciferase activity reporter, suggesting its influence on the protein expression of SLC35F2. The mRNA level of SLC35F2 was increased in MIBC compared with NMIBC. Furthermore, in immunohistochemical staining, we observed a strong intensity of SLC35F2 in single tumour cells and in the border cells of solid tumour areas with an atypical accumulation around the nucleus, especially in the MIBC. This suggests that SLC35F2 might be highly expressed in aggressive and invasive tumour cells. Moreover, knockdown of SLC35F2 repressed the growth of bladder cancer cells in the monolayer and spheroid model and suppressed migration and invasion of bladder cancer cells. In conclusion, we suggest that SLC35F2 is involved in bladder cancer progression and might provide a new therapeutic approach, for example, by the anti-cancer drug YM155, a cargo of the SLC35F2 transporter.

## 1. Introduction

Bladder cancer (BC) is one of the most common cancers worldwide, with approximately 430,000 new cases per year [1]. The main causes of BC are occupational exposure to urothelial-related carcinogens and smoking. Moreover, men are more frequently affected than women, with an approximate ratio of 3:1 [2,3]. BC originates from the urothelial layer of the bladder. At first diagnosis, approximately 75% of BCs are still in a non-muscle-invasive state (non-muscle-invasive BC, NMIBC), while 25% are muscle-invasive BC (MIBC). NMIBC has a high recurrence rate and approximately 15% progress to a muscle-invasive state with poor prognosis [4,5,6]. The molecular mechanisms of BC pathogenesis and progression are insufficiently understood. Their understanding and the identification of target genes related to BC carcinogenesis are thus urgently required in order to provide new therapeutic opportunities.

Solute carrier family 35 member F2 (SLC35F2) belongs to the SLC35 family [7]. Thereof, most members are involved in the transport of nucleotide sugars and seem to be predominately localized in the endoplasmic reticulum (ER) and/or Golgi apparatus. SLC35F2 is not well studied, and its substrates and localisation, although the latter has been suggested to be located at the outer cell membrane, remain unclear [7,8,9,10]. In addition, the functional role of SLC35F2 in several diseases and cancer types has not been analysed intensively so far. SLC35F2 was first described in ataxia-telangiectasia [11] and some studies have shown that SLC35F2 is involved in the development of various cancer types. Patients with non-small cell lung carcinoma (NSCLC) showed high expression levels of SLC35F2, which correlated with pathological staging and suggested the use of SLC35F2 as a prognostic marker in NSCLC [12]. Knockdown of SLC35F2 in H1299 lung cancer cells reduced their proliferation, migration, and invasion, likewise supporting the role of SLC35F2 as an oncogene in lung cancer [13]. In addition, high levels of SLC35F2 were positively correlated with progression of papillary thyroid carcinoma (PTC) and knockdown of SLC35F2 suppressed the malignant phenotype of PTC cells in vitro and in vivo [14]. Winter et al. showed that SLC35F2 transported the anti-cancer drug YM155 into KBM7 cells, which thereafter induced DNA damage and resulted in cell death. They also showed that disruption of SLC35F2 expression promoted drug resistance in YM155, proposing an important role of SLC35F2 in the mechanisms of drug resistance [15]. Moreover, in prostate cancer, SLC35F2 seems to play a role, as it was highly expressed, directly upregulated by the androgen receptor (AR) activation and crucially involved in the sensitivity of prostate cancer cells to YM155 [16]. One recent publication showed that SLC35F2 is highly expressed in BC tissues and can promote BC progression [17]. Therefore, understanding the molecular mechanism of SLC35F2 in the progression of cancer might provide new therapeutic opportunities against BC.

In this study, we identified frequent mutations in the 5′ and 3′ untranslated region (UTR) of various tumour-associated genes, including SLC35F2, from whole-exome and UTR sequencing and focused on the investigation of the functional role of SLC35F2 in BC.

## 2. Materials and Methods

### 2.1. Patients and Samples

Tumour tissue samples were obtained from patients subjected to transurethral resection of the bladder or radical cystectomy at the Department of Urology, Jena University Hospital, Germany. Samples for immunohistochemistry were formalin-fixed; embedded in paraffin for pathological examination; and stored at the Institute of Pathology, Jena University Hospital, Jena, Germany, until use. Frozen tissue from the same patients used for sequencing and mRNA expression analysis originated from the biobank of the above-mentioned Department of Urology. The baseline characteristics (age, sex, tumour stage, and grade) of the subjects are summarized in Table 1. All patients gave their written informed consent to provide residual tissue for research. The biobank and the presented study were approved by the institutional ethical committee of Jena University Hospital (No. 3657-01/13; No. 4213-09/14). The study was conducted in accordance with the Declaration of Helsinki.

### 2.2. Whole Exome and UTR Sequencing

Cryosections of tumours were stained with haematoxylin and eosin (H & E) to estimate tumour cell content and to select areas for microdissection by a pathologist. In order to obtain tumour cell contents of over 50%, we manually dissected up to 30 cresyl violet-stained 12 µm cryosections per sample by scraping with scalpel under an inverse microscope. DNA was isolated by phenol-chloroform extraction. Library preparation was done for matched blood as well as NMIBC and MIBC samples by Agilent SureSelect QXT Target Enrichment for Illumina Multiplexed Sequencing protocol with SureSelect XT Human All Exon V5 + UTRs library (Agilent Technologies, Waldbronn, Germany) according to the manufacturer’s protocol. All libraries were sequenced on Illumina HiSeq platform using the 2 × 100 bp paired-end high output mode and v3 chemistry by ATLAS Biolabs, Germany. Bioinformatics and detection of somatic mutations were done by the Bioinformatics/High-Throughput Analysis group, Faculty of Mathematics and Computer Science, Friedrich Schiller University Jena, Germany. Details of bioinformatics material and methods are available as Appendix A. The cut-off for considered mutations was 0.1 for allele frequency and 150 for quality score. Mutations that affected UTRs in at least three different patients were checked for incorrect sequencing caused by long nucleotide repeats and for annotated UTR variations in the reference genome hg38 and were excluded from the analysis. Whole sequencing data of all 11 patients were deposited in the European Nucleotide Archive (ENA) at EMBL-EBI under accession number PRJEB41286 (https://www.ebi.ac.uk/ena/browser/view/PRJEB41286).

### 2.3. RNA Extraction

For the extraction of total RNA from tumour tissues and normal healthy ureters, a combined protocol between Triazol reagent (Thermo Fisher Scientific, Waltham, MA, USA) and NucleoSpin RNA XS (Macherey-Nagel, Düren, Germany) was used. Micro-dissected tumour samples with at least 75% tumour cell content and healthy ureters were lysed in 1 ml Triazol, followed by phase separation, as described in the Triazol user guide. The aqueous phase (RNA) was thoroughly mixed with an equal volume of 70% ethanol, followed by RNA purification according to the manufacturer’s instructions (NucleoSpin RNA XS).

The RNA from cells was isolated using Triazol reagent according to the manufacturer’s protocol. RNA was quantified using Qubit 3.0 Fluorometer (Invitrogen, Thermo Fisher Scientific, Waltham, MA, USA) and RNA quality was analysed by Agilent 2200 TapeStation (Agilent, Waldbronn, Germany) instrument according to the manufacturer’s protocols. The extracted RNA was stored at −80 °C.

### 2.4. cDNA Synthesis and Reverse Transcription Quantitative PCR (RT-qPCR)

Here, 2 μg of total RNA was reverse transcribed into cDNA using GoScript Reverse Transcription System (Promega, Mannheim, Germany) according to the manufacturer’s protocol. The cDNA was diluted 1:8 with sterile water and stored at −20 °C. RT-qPCR analysis was performed using LightCycler 480 SYBR Green I master mix and the LightCycler 480 instrument (Roche Applied Science, Penzberg, Germany). The Cp value was extracted using the second derivative maximum method from the LightCycler software. The relative expression to the housekeeping genes RPS13 and RPS23 was calculated by the ΔΔCp method, including primer efficiency in the calculation [18]. Sequences of primers are listed in the Appendix A and the experiments were repeated at least three times.

### 2.5. Immunohistochemistry and H & E Staining

Here, 4 µm sections from formalin-fixed paraffin-embedded (FFPE) tumour samples were immunohistochemically stained with Dako REAL EnVision Detection System K5007 (Agilent, Germany) according to the manufacturer’s protocol against SLC35F2 (anti-SLC35F2 1:100, HPA048185, Atlas antibodies, Bromma, Sweden). Heat-induced epitope retrieval was done in Tris-EDTA Buffer (10 mM Tris Base, 1 mM EDTA, 0.05% Tween20, pH 9.0) at 98 °C for 20 min. A consecutive section was H & E-stained. Immunohistochemical staining was assessed by intensity (no staining (−), weak (+), moderate (++), strong (+++)) of tumour cells. The results were validated and confirmed by a uropathologist.

### 2.6. Cell Culture

T24, TCCsup, RT112, and RT4 cell lines were cultured in RPMI 1640 medium with Glutamax (Thermo Scientific) supplemented with 10% fetal bovine serum (FBS), penicillin (100 U/mL), and streptomycin (100 µg/mL). Cal29 cell line was cultured in Dulbecco’s modified Eagle medium (DMEM) supplemented with 10% FBS, penicillin (100 U/mL), and streptomycin (100 µg/mL). All cells were incubated in a humidified atmosphere at 37 °C with 5% CO_2_. All cell lines were commercially obtained in the 1990s. Short tandem repeat (STR) analysis was last done in June 2018 by the Institute of Forensic Medicine, Jena University Hospital, Germany, using PowerPlex 16 System (Promega, Germany) for amplification and capillary gel electrophoresis for analysing allele profiles. Profiles were checked by German Collection of Microorganisms and Cell Cultures (DSMZ) online STR analysis.

### 2.7. Western Blotting

For the preparation of protein extract, cells were lysed in NETN buffer (100 mM NaCl, 1 mM EDTA, 20 mM Tris/HCl pH 8.0, 0.5% NP-40) supplemented with protease inhibitors (10 μg/mL Leupeptin and 1 mM phenylmethylsulfonyl fluoride (PMSF)) on ice for 10 min, followed by three cycles of freezing (liquid nitrogen) and thawing (37 °C). The cell debris was removed by centrifugation (15,000 rpm, 15 min, 4 °C). The protein extracts were separated and blotted on a Polyvinylidene difluoride (PVDF) membrane by 10% SDS-PAGE. The following antibodies were used: anti-SLC35F2 (AV43971, Sigma-Aldrich, St. Louis, Missouri, MO, USA) and anti-α-Tubulin (sc-5286, Santa Cruz, Dallas, TX, USA). Horseradish peroxidase-conjugated anti-mouse IgG-HRP (sc-516102, Santa Cruz, Germany) or anti-rabbit IgG-HRP (sc-2370, Santa Cruz, Germany) were used as secondary antibodies. The membrane was incubated with chemiluminescence (ECL) reagent (GE Healthcare, Solingen, Germany) and the produced light emission was detected by GBox Chemi XX6 (Syngene, Cambridge, GB).

### 2.8. Luciferase Reporter Gene Assay

Wildtype and mutated 5´UTR (C/G, position 269, NM_017515.4) of SLC35F2 were synthesized and cloned between EcoRV and BstEII restriction sites upstream of the NLucP luciferase gene in the pmirNanoGLO vector (Promega, Germany) by Eurofins Genomics, Germany. Wildtype and double mutated 3’UTR (C/T at position 1935 and C/G at position 1632, NM_017515.4) of SLC35F2 were cloned between NheI and SalI restriction sites downstream of the NLucP by Eurofins Genomics, Germany. The single mutations of the 3′UTR were created using the 3′UTR wildtype construct as template using Q5 Site-Directed Mutagenesis Kit (New England Biolabs, United Kingdom), according to the manufacturer’s instructions, utilizing specific back-to-back mutagenesis primers (Appendix A). The successful mutated UTRs were verified by sequencing and re-subcloned in a new pmirNanoGLO vector to avoid any mutations in the backbone during the mutagenesis method.

T24 and TCCsup cells were seeded on 24-well plates and transfected after 24 h with luciferase reporter gene vectors using jetPrime transfection reagent (Polyplus, New York, NY, USA) according to the manufacturer’s protocol. After 24 h, the luciferase activity was measured using the Dual-Glo Luciferase Assay System (N1630, Promega, Mannheim, Germany) on the Tecan Infinite M200 Pro reader (Tecan, Männedorf, Swizerland). Moreover, RNA was extracted in order to analyze the luciferase mRNA level. The NLucP luciferase activity and mRNA level were normalized to Luc2 luciferase to eliminate variations in the transfection efficiency (NLucP/Luc2). Fold change was calculated to emphasize the effects of a single mutation on luciferase activity and mRNA level. All experiments were repeated at least three times.

### 2.9. Growth and Spheroid Formation Assays

For growth assay, T24 and Cal29 cells were seeded on six-well culture plates. After 24 h, the cells were transfected with 40 μM pooled siSLC35F2 (SI04178020, SI04345544, SI04267613 Qiagen, Hilden, Germany) using INTERFERin reagent following the manufacturer’s protocol (Polyplus, New York, NY, USA). Allstars negative control siRNA (SI0365031, Qiagen, Germany) was used as a negative control. Six days post-transfection, cells were washed with phosphate-buffered saline (PBS), fixed with 2% glutaraldehyde for 10 min, and stained with 0.1% crystal violet for 30 min. Cells were gently washed with distilled water, dried overnight, and thereafter solubilized with lysing solution (0.1 M sodium citrate, 50% ethanol, pH 4.2) for 30 min with gentle shaking. The absorbance was measured on Infinite M200 Pro reader at 590 nm (Tecan, Austria). For spheroid formation, T24 cells were seeded on six-well culture plates and transfected after 24 h with 40 μM pooled siSLC35F2 as described above. Twenty-four hours post-transfection, the cells were washed once with PBS, trypsinated, and collected at 300 rpm for 10 min. The cell pellet was resuspended with fresh growth medium and 1000 cells were seeded per well of cell-repellent surface 96-well microplate (Greiner Bio-one, Kemsmünster, Austria). Afterwards, the plate was centrifuged at 500 rpm for 3 min and pictures were taken using Cell observer (Cellobserver Z1 with Colibri-7, Zeiss, Oberkochen, Germany). The experiments were repeated at least three times. The software MATLAB R20019a and AnaSP were used to extract the volume from spheroid formation experiments.

### 2.10. xCELLigence Real-Time Cellular Analysis

Migration and invasion of cells were analysed using the xCELLigence RTCA System (ACEA Bioscience, San Diego, CA, USA) according to the manufacturer’s protocol (Cell Migration and Invasion protocol). The cells were transfected with 40 μM pooled siSLC35F2 on a six-well plate as previously described. Allstars negative control was used as a negative control (Qiagen, Hilden, Germany). Seventy-two hours post-transfection, the transfected cells were trypsinated and resuspended in serum-free medium. For migration, 20,000 cells in 100 μL were added to the upper chamber of CIM-plates. In the case of invasion, 40,000 cells in 100 μL were added to the upper chamber wells, pre-coated with 1:12 or 1:40 diluted Matrigel (catalogue no. 354230, Corning, New York, NY, USA). In the lower chamber, medium with 10% serum was added. All experiments were repeated at least three times.

### 2.11. Data and Statistical Analysis

Statistical analysis was performed using the software IBM SPSS Statistics Version 25 by combined biological replicates and the data are expressed as mean ± standard deviation. The non-parametric Mann–Whitney U-test was performed to compare two groups. A 95% confidence interval (*p*-value < 0.05) was considered as statistically significant (*).

## 3. Results

### 3.1. The Oncogene SLC35F2 Carries Frequent UTR Mutations in Bladder Cancer Samples

Frozen tissue for exome and UTR sequencing of the NMIBC and the matched metachronous MIBC was available from eight patients with progressive disease. From an additional three patients with progressive disease, only the MIBC was available for sequencing (Table 1). We detected genes significantly mutated in The Cancer Genome Atlas study of bladder cancer 2017 (exome sequencing of 412 MIBC) with similar frequencies of mutations in coding sequences in our sample cohort [19]. We also detected well-known mutated genes from the Uromol study, analysing 460 NMIBC by RNA sequencing [20] (Appendix A).

Furthermore, exome-wide sequencing with a focus on untranslated regions revealed frequent mutations in the regulatory 5′ and 3′ regions of mRNA transcripts of genes. One of these genes was SLC35F2, carrying UTR mutations in the NMIBC of three patients. The allele frequency was 0.26 for the 3′ UTR mutations at position 1935 (C/T) and 1632 (C/G) NM_017515.4, respectively, and 0.3 for the 5′ UTR mutations at position 269 (C/G) NM_017515.4.

### 3.2. SLC35F2 Is Highly Expressed in Patients with MIBC

We analysed the mRNA expression of SLC35F2 by quantitative PCR in the same tissue samples used for sequencing as well as in three healthy ureters and five BC cell lines. The expression of SLC35F2 was significantly increased in MIBC compared with NMIBC and healthy ureters (*p* < 0.05, Figure 1A). The mRNA level of SLC35F2 in BC cell lines (Cal-29, RT4, T24, TCC-sup, RT112) varies relative to the housekeeping genes in the range of 0.04 and 0.06 (Figure 1B). We selected Cal29 with higher mRNA expression of SLC35F2 and T24 with lower expression of SLC35F2 for invasion and migration analysis.

Furthermore, we analysed the protein expression of SLC35F2 in corresponding FFPE tumour tissues. Only for patient 3 was FFPE tissue of the NMIBC not available. Typical urothelial layer without dysplasia, if present in the sample, was uniformly weakly stained (rarely not stained) in the cytoplasm (Figure 2A). We detected a cytoplasmic weak to moderate staining in atypical urothelial layers and a weak to strong staining of tumour cell areas (Figure 2, Appendix A). In general, no substantial differences in average staining intensity were seen between non-muscle invasive and muscle-invasive tumour cohorts; however, the staining intensity of MIBC tended to be increased (mainly moderate staining) compared with the NMIBC (mainly weak stained) (Appendix A). Comparing matched NMIBC and MIBC of single patients, some tumour areas in MIBC showed an increased cytoplasmic staining in the border of tumour complexes or in single tumour cells (Figure 2C). Partially, this increased staining was selectively close to the nuclear membrane. This selective staining was increased in the MIBC compared with the matched NMIBC samples, especially in patients 2 and 5–8 (Appendix A).

### 3.3. Mutation in the 3´UTR of SLC35F2 Increases Luciferase Activity

In order to analyse whether the UTR mutations in SLC35F2 play a role in the regulation of gene expression, we analysed the effect of UTR mutations by luciferase reporter gene assay. We were able to clone the whole wildtype or mutated 5´UTR sequences (421 bp) upstream and wildtype and mutated 3′UTR sequences (1624 bp) downstream of the luciferase gene in the pmirNanoGLO vector (Figure 3A) to mimic the biological function of the whole UTRs. We observed a significant increase in luciferase activity of 20% by the double mutated 3′UTR (C/T at position 1935 and G/C at position 1632 C/G, NM_017515.4) as well as a 20% increase in activity by the single mutation C/T compared with the wildtype sequence (*p* < 0.05) in T24 (Figure 3B). In TCCsup, similar results were observed. The other two single mutations (C/G in 5′UTR, C/G 3′UTR) did not influence the luciferase activity (Figure 3B). To study whether the UTR mutations affect transcription or translation, we additionally analysed the luciferase mRNA level, but did not observe any changes in the mutated constructs compared with the wildtype (Figure 3C). In conclusion, the results show that one of the mutations in the 3´UTR increases luciferase activity in vitro.

### 3.4. SLC35F2 Promotes Growth of Bladder Cancer Cells in Monolayer and Spheroid Model

In line with the observation from the expression analysis of SLC35F2 in tumour samples, we investigated the influence of SLC35F2 knockdown on BCa cells T24 and Cal29 by transfection with siRNA against SLC35F2. SLC35F2 mRNA and protein levels were significantly reduced compared with siControl (Figure 4A,B, Appendix A). We analysed the effect of SLC35F2 knockdown on the growth of T24 and Cal29 cells. Crystal violet assay at day 6 after transfection showed that knockdown of SLC35F2 significantly repressed the growth of T24 and Cal29 (Figure 4C, *p* < 0.05). The spheroid model mimics better tumour growth in vivo in the aspect of complexity compared with the monolayer model. For this reason, we investigated the influence of SLC35F2 knockdown on spheroids. We observed a significant reduction of spheroid volume after siSLC35F2 knockdown after six days compared with control siRNA transfected cells (Figure 4D). We were not able to analyse the effect of siSLC35F2 on Cal29 spheroids because the spheroids died after 3 days. Overall, the data show that knockdown of SLC35F2 reduces the growth of BC cells.

### 3.5. Knockdown of SLC35F2 Inhibits Migration and Invasion of BC

We investigated the influence of SLC35F2 on migration and invasion in BC cells in real-time using the xCELLigence RTCA System after transfection with siSLC35F2. The knockdown of SLC35F2 strongly inhibited the migration of T24 and Cal29 compared with siControl (Figure 5A,B). Knockdown of SLC35F2 repressed the invasion of T24 cells through 1:12 diluted matrigel by the same system (Figure 5C). Under these conditions, the invasive capacity of Cal29 compared with T24 was very low. Therefore, we analyzed the invasiveness of Cal29 cells through 1:40 diluted matrigel and we observed a significant repression of invasion by siSLC35F2. Similar results were observed for three biological replicates, respectively (Appendix A). Our data show that SLC35F2 knockdown inhibits migration and invasion of BC cells.

## 4. Discussion

The molecular mechanisms of BC pathogenesis are poorly understood. Studies from DNA and RNA sequencing analysis suggest that BC shows high mutational heterogeneity and new mutations are accumulated during carcinogenesis and progression [21,22,23]. From whole-exome and UTR sequencing analysis of NMIBC and MIBC from 11 patients, we identified frequent mutations in the coding sequence as well as in 5´ and 3´ UTR of different tumour-associated genes. By comparing mutation databases (TCGA) and sequencing studies (Uromol) for bladder cancer, we were able to confirm the different patterns of coding mutations between NMIBC and MIBC in our samples [19,20]. Furthermore, we observed that the SLC35F2 transporter contains point mutations in the 5′ and 3′ UTR in NMIBC of three patients. The ratio of common mutation between NMIBC and matched MIBC of these three patients was low (0.6%, 0.9%, and 12.6% compared with the other patients with up to 65.8%), indicating that the sequenced MIBC was not directly derived from the sequenced NMIBC (Table 1). Nevertheless, in this study, we focused on deciphering the functional role of SLC35F2 in BC and on the probable influence of the mutations on the expression of SLC35F2.

It has been reported that many members of the SLC superfamily with at least 362 transporters were mutated and overexpressed in many diseases and cancer [7,24], suggesting their crucial importance in the respective diseases. The functional role and molecular mechanisms of SLC35F2 in cancer development have not yet been comprehensively investigated. It has been shown that SLC35F2 is highly expressed in NSCLC, PTC, prostate cancer, and BC, and the authors suggest that high SLC35F2 expression is correlated with cancer progression [12,14,16,17]. Our data suggest that SLC35F2 is higher expressed in MIBC as compared with matched NMIBC, indicating that SLC35F2 might be involved in BC progression. This is consistent with observations from Chen et al., who suggested the high SLC35F2 level to be correlated with tumour stage and invasiveness of BC [17]. Here, we show an increased staining (moderate to strong) of SLC35F2 in single tumour cells and in the cells of the border of solid tumour areas in MIBC compared with matched NMIBC samples, suggesting that these cells might be responsible for the invasiveness and aggressiveness of BC. In this group of tumour cells, we also detected some cells with an atypically strong and selective staining of SLC35F2 around the nucleus in MIBC. We hypothesize that this increased protein accumulation of SLC35F2 around the nucleus is probably at the membranes of the ER or Golgi apparatus, as described for the SLC35 family by Parker et al. [25]. This might help cancer cells to increase their own metabolism and might lead to a subsequently increased cancer activity. Frezza et al. showed that cancer metabolism plays a crucial role in proliferation, migration, invasion, and metastatic spread of cancer [26]. In line with this, a high level of the SLC35F2 transporter might probably contribute to tumour progression through the increase of cancer metabolism. In addition, some studies show that stabilization of β-catenin and androgen receptor signaling can increase the expression of SLC35F2, providing new insights into the understanding of the regulation of this gene and potential targets for therapy [16,27,28]. Interestingly, the anti-cancer drug YM155 is a cargo of the SLC35F2 transporter, which has been reported to be involved in the drug resistance of YM155. It is assumed that high levels of SLC35F2 are associated with an increase of YM155 sensitivity in cancer therapy [15,16,29]. Other studies demonstrated that YM155 inhibited the growth of various cancer cell lines and showed anti-tumour activity in xenograft models, including BC [30,31]. We believe that understanding the physiological role of SLC35F2 in BC could provide new therapeutic opportunities with the anti-cancer drug YM155 being beneficial in patients with high levels of SLC35F2. Further experiments are needed to clarify the link between SLC35F2 and YM155 in BC.

Next, we focused on understanding the role of detected UTR mutations of SLC35F2 because it is well known that UTRs of mRNA have a crucial role in the regulation of gene expression [32]. Only the C/T mutation in 3′UTR showed a significant increase of luciferase activity; however, no change in the luciferase mRNA level was detected. This suggests that the mutation might increase the protein expression of SLC35F2 at a translation and not at an mRNA level. This supports the idea that the C/T mutation in 3′UTR might have an oncogenic effect because it might increase the expression of the oncogene gene SLC35F2, providing some hypothesis that UTR mutation might be involved in cancer progression. It is conceivable that the mutation might disrupt or create binding sites for miRNA, leading to an increased expression of SLC35F2. Nevertheless, we did not detect an increased average staining in the tumour with the 3′UTR mutation compared with other NMIBC, which suggests that this 3’UTR mutation may play a minor role in the biology of SLC35F2.

We emphasize that cancer progression and carcinogenesis may result from multiple mutations and not because of a single one. Further work is necessary to substantiate the role of UTR mutations in BC development, with the insertion of these mutations into different BC cell lines by CRISPR/cas and subsequent analysis of their functional role being one approach.

As mentioned, SLC35F2 acts as an oncogene in various tumour entities. Here, we showed that knockdown of SLC35F2 significantly repressed the growth of T24 and Cal29 cells in a monolayer model, as well as of T24 spheroids, suggesting that SLC35F2 promotes the growth of BC cells. Furthermore, we showed that knockdown of SLC35F2 repressed the migration and invasion in T24 and Cal29 cells, suggesting that SLC35F2 contributes to the migration and invasion of bladder cancer cells. Overall, our data confirm that SLC35F2 is an oncogene in BC cells.

## 5. Conclusions

SLC35F2, a transporter of the anti-cancer drug YM155, has been strongly associated with cancer progression. We identified various tumour-associated genes, including SLC35F2, to be frequently mutated in our patient samples. We showed that one mutation in 3′UTR might increase the protein expression of SLC35F2, providing indications that UTR mutations might be involved in gene regulation and, subsequently, in cancer progression. Our results indicate that SLC35F2 expression is increased in patients with MIBC compared with matched NMIBC and provide evidence that SLC35F2 promotes growth, migration, and invasion in BC cells, suggesting its oncogenic role in BC. We suggest that SLC35F2 is involved in BC progression and that understanding of the molecular mechanism of SLC35F2 in carcinogenesis might provide new therapeutic opportunities in BC. Herein, we explicitly refer to the SLC35F2 cargo YM155, which is currently tested in clinical trials for treatments of different cancer types.

## Figures and Tables

**Figure 1 cells-10-00080-f001:**
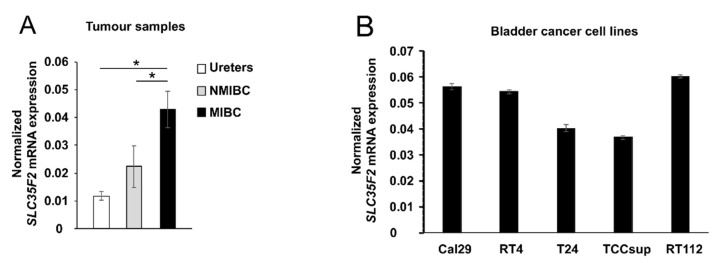
mRNA expression analysis of SLC35F2 in tumour samples and bladder cancer cell lines. (**A**) SLC35F2 mRNA expression of eight matched non-muscle-invasive bladder cancer (NMIBC) and MIBC patients; (**B**) SLC35F2 mRNA expression in bladder cancer cell lines. Expression was normalized to housekeeping genes RPS13 and RPS23. The data are expressed as mean ± standard deviation and statistical analysis was performed by non-parametric Mann–Whitney U-test with *****
*p*-value < 0.05.

**Figure 2 cells-10-00080-f002:**
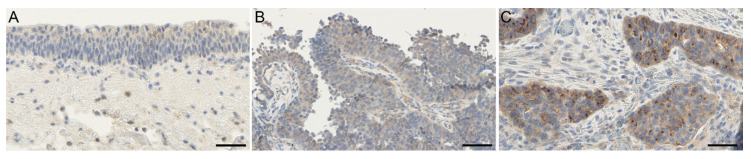
Immunohistochemical staining of SLC35F2 in patient 7. (**A**) Not/weak stained typical urothelium; (**B**) NMIBC, weak stained; (**C**) MIBC, moderate to strong stained. SLC35F2 is weak to moderate and uniformly stained in urothelium and in the majority of NMIBC. Increased staining of SLC35F2 was detected mainly around the nucleus of tumour cells in MIBC compared with NMIBC and typical urothelium. Scale bar = 50 µm.

**Figure 3 cells-10-00080-f003:**
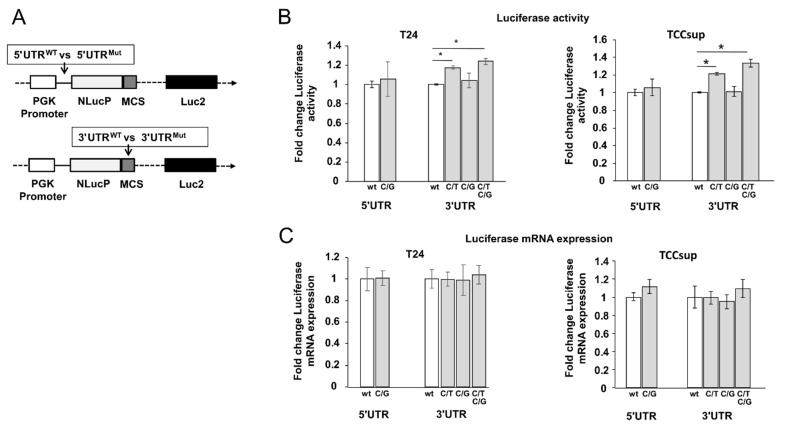
Untranslated region (UTR)-mutation analysis of SLC35F2 by luciferase reporter assay. (**A**) Schematic view of the whole UTR sequencing cloned in luciferase reporter. The wildtype and mutated 5′UTR sequences were cloned upstream, while the wildtype and mutated 3′UTR were cloned downstream of the NLucP reporter. The bladder cancer cell lines T24 (*n* = 4) and TCCsup (*n* = 4) were transfected with the luciferase plasmids and after 24 h luciferase activity (**B**) and mRNA (**C**) were analysed. The Luc2 luciferase was used for normalisation to eliminate variations from transfection. The data are expressed as mean ± standard deviation and statistical analysis was performed by non-parametric Mann–Whitney U-test with * *p*-value < 0.05. Values obtained from wild type sequences were set arbitrarily as 1. PGK: phosphoglycerate kinase, MCS: Multiple Cloning Site.

**Figure 4 cells-10-00080-f004:**
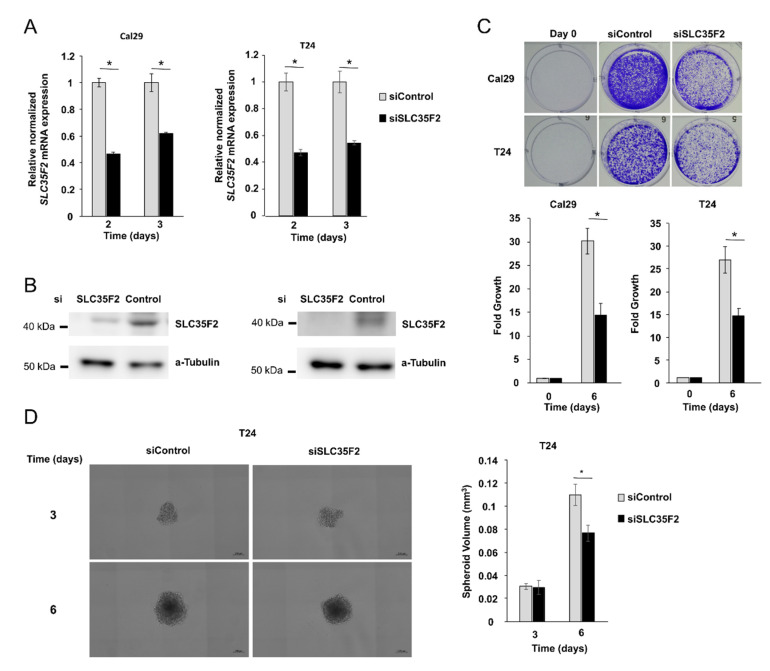
SLC35F2-dependent growth in Cal29 and T24 bladder cancer cells. Knockdown of SLC35F2 by siRNA leads to a decrease in mRNA expression (*n* = 4) (**A**) as well as protein expression (**B**) (for full image, see Appendix A) in Cal29 and T24. Cell growth is significantly reduced in SLC35F2 knockdown cells, analysed by crystal violet staining assay (*n* = 4) (**C**). Spheroid growth of T24 is also significantly reduced after six days (*n* = 6) (**D**). * *p* < 0.05 Mann–Whitney-U test.

**Figure 5 cells-10-00080-f005:**
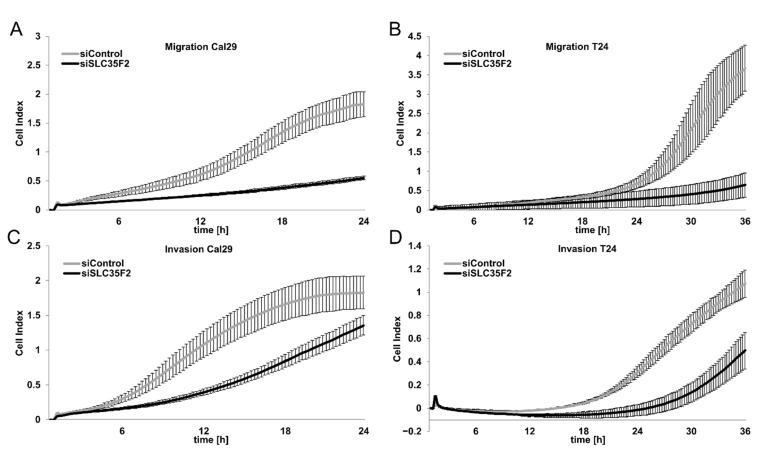
SLC35F5-dependent migration and invasion of bladder cancer cells. Migration of Cal29 (**A**) and T24 (**B**) is strongly reduced after siRNA knockdown of SLC35F2. The invasiveness of T24 (**C**) and Cal29 (**D**) was significantly reduced using 1:12 and 1:40 diluted Matrigel, respectively. Here, 20,000 cells were used for migration analysis and 40,000 cells were used for invasion analysis. The cell index reflects the amount of cells that pass the membrane between the upper and lower chamber, analysed by xCelligence real-time system.

**Table 1 cells-10-00080-t001:** Characteristics of patients and tumour samples.

Patient			Primary NMIBC		NMIBC Recurrence				MIBC	Common SNV in Seq. Samples
No	Age	Sex	TNM-T/Grading	RFS [m]	TNM-T/Grading	RFS [m]	TNM-T/Grading	PFS [m]	TNM-T/Grading	
1	67	m	pTa HG	59	**pT1 HG**			15	**pT4a HG**	**13.9%**
2	79	w	**pT1 HG**	15	**pTa LG**	23	3x pTa HG	5	**pT2b HG**	**65.8%**
3	69	m	**pTa HG**					10	**pT4a HG**	**22.8%**
4	78	w	**pT1 HG**	30	pTa HG			12	**pT4a HG**	**28.1%**
5	65	m	**pTa HG**	13	PUNLMP			21	**pT2a HG**	**2.2%**
6	71	m	**pT1 HG**	10	pTa LG	41	pT1 LG	23	**pT2a HG**	**12.6%**
7	74	m	**pTa HG**					42	**pT3a HG**	**0.9%**
8	73	m	**pTa LG**					62	**pT2 HG**	**0.6%**
9	63	m	pTa HG	29	pTa HG			10	**pT2a HG**	
10	76	m	pTa HG					16	**pT3b HG**	
11	76	m	pTa HG	14	pTa HG	5	pTa LG	28	**pT4a HG**	

Bold: exome-wide sequenced tumors; NMIBC, non-muscle-invasive bladder cancer; MIBC, muscle-invasive bladder cancer; RFS, recurrence free survival; PFS, progression free survival; TNM-T, classification of size and extension of malignant tumors, LG, low grade; HG, high grade; PUNLMP, papillary urothelial neoplasm of low malignant potential; com. SNV, common single nucleotide variations.

## Data Availability

Data is contained within the article or Appendix A. Whole sequencing data of all 11 patients were deposited in the European Nucleotide Archive (ENA) at EMBL-EBI under accession number PRJEB41286 (https://www.ebi.ac.uk/ena/browser/view/PRJEB41286). The whole scans of IHC slides are available on request from the corresponding author.

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
