# Peer review of "SLC35F2, a Transporter Sporadically Mutated in the Untranslated Region, Promotes Growth, Migration, and Invasion of Bladder Cancer Cells"

_cells, 2021, doi:10.3390/cells10010080_

Round 1

Reviewer 1 Report

Kotolloshi and colleagues' manuscript describes the finding of a mutation in the 3'UTR region of SLC35F2, which the authors linked to increased tumorigenesis in bladder cancer. This descriptive article is well written and very informative, but the authors' conclusions are not always supported by the data presented. Furthermore, only three patients analyzed showed UTR mutations. Below is a list of criticisms.

Major criticism:

  1. Figure 2: The immunostaining of SLC35F2 lacks proper statistical quantification. Moreover, panel A derives from patient 2, whereas the tissues in panels B and C belong to patient 5. This isn't very clear. If the authors want to compare tissues from different patients, then a statistical analysis of SLC35F2 staining intensity in all patients must be performed and shown. If the authors wish to show a purely qualitative figure, then at least tissues from the same patient should be used in all three panels.
  2. Lines 239-241. "This selective staining was significantly increased…." Supplementary figure S1 does not show any statistical analysis. Therefore this sentence is incorrect.

Minor criticisms:

  1. Table 1: the design of table 1 is not straightforward. If patient, primary NMIBC, NMIBC recurrence, and MIBC are the main groups/categories, they should be better defined. Perhaps vertical lines will help better organize the table. For example, the authors might consider inserting a vertical line between sex and TNM-T/grading and distinguishing the four main groups.
  2. Line 67. A full stop is missing at the end of the paragraph. 
  3. Line 274. Here the authors refer to supplementary figure s2 and not s3 as written.

Reviewer 2 Report

Summary

In their paper, Kotolloshi et al studied the role of SLC35F2 in bladder cancer (NMIBC and MIBC). Authors performed exome and UTR sequencing on NMIBC and matched MIBC tumor samples, identified a list of mutations in well-known bladder cancer genes, and found several UTR mutations, including some in the SLC35F2 gene from 3 NMIBC samples (but not in the corresponding MIBCs!). Also, based on the results of a luciferase-reporter assay, authors claimed that one of these UTR mutations might “increase the protein expression of SLC35F2”.

Authors also showed that SLC35F2 mRNA and protein levels were higher in MIBC as compared to NMIBC and normal ureters. SLC35F2 was further studied in vitro: authors found that siRNA mediated downregulation of SLC35F2 in two BC cell lines impacted their capacity to grow, as well as their in-vitro migration and invasion potential.

In conclusion, authors showed that SLC25F2 is often upregulated in muscle-invasive bladder cancer and could contribute to the process tumor progression.

Major points

  1. The involvement of UTR mutations in the regulation of SLC35F2 function is questionable. These mutations were only identified in NMIBCs and not in the matched MIBCs, and their effect on SLC35F2 translation was not studied. In order to justify the inclusion of the "UTR mutation" story in the paper, authors should at a minimum perform a series of experiments aimed at assessing the relationship between mRNA and protein levels of different constructs w/ or w/o these UTR mutations. Also, does the NMIBC IHC slide from the patient carrying this C/T 3’ UTR mutation show stronger SLC35F2 intensity compared to other NMIBC slides?
  2. The claim that IHC stainings revealed a peri-nuclear (Golgi/ER) SLC35F2 staining is unsubstantiated. The hypothesis is reasonable, but authors should further study the localization of the protein at higher resolution (together with Golgi/ER markers). Does SLC35F2 show a peri-nuclear, Golgi, or ER localization in BC cell lines (by confocal microscopy)?
  3. Figure 1B is not providing clear insights. Unlike what stated by the authors, different BC cell lines express different levels of SLC35F2, ranging between ~0.035 and ~0.06. Analyzing SLC35F2 mRNA levels in one or two immortalized human urothelial cell lines could be highly informative. Also, does over-expression of SLC35F2 in an immortalized human urothelial cells increase their growth, migration, and invasion potential?
  4. In figure S2, what is gene X? Why does a siRNA directed against gene X down-regulate all SLC35F2 bands in Cal29 cells as compared to a siCONTROL? I honestly find these Western Blots very confusing. Is the silencing really specific?

Minor

  1. the sentence at line 35 <an approximate ratio of 3:1 being mainly caused by occupational exposure to urothelial-related carcinogens and smoking> is misleading since sex is believed to play a role in BC, together with environmental exposures.
  2. the sentence at line 37 is inaccurate, since 25% of bladder tumors are muscle-invasive, but not all muscle-invasive bladder tumors are metastatic at diagnosis
  3. It would be useful to list SLC35F2 IHC signal quantification results below each panel in Fig S1
  4. In figure 1A, is it possible that the differences in mRNA levels are mainly driven by a different percentage of cells of epithelial origin in the input samples? If so, authors should discuss this possibility
  5. In Figure 1B, comparing mRNA levels from tumor samples and cultured cells seems inappropriate.
  6. what drives SLC35F2 mRNA and protein over expression in MIBC? While this likely goes beyond the scope of this study, authors may want to comment on that.
  7. typo in line 315: "focussed"

Author Response

Please see attachement.

Round 2

Reviewer 2 Report

  1. My major point is about the interpretation of the data about the C/T 3'UTR mutation in BC. The luciferase assay was an indirect way of assessing protein levels using a reporter. A transfection- and reporter-based system can only SUGGEST but DOES NOT IMPLY an effect on the expression levels of the protein of interest. Therefore, from my point of view authors are over-interpreting their findings. Moreover, the admission that the C/T 3'UTR mutation-carrying tumor did not show an increased staining of SLC35F2 compared to other NMIBC tumors suggests that this 3'UTR mutation may play a minor role in the biology of SLC35F2. Unfortunately, authors did not consider running further experiments to better address this point.
  2. About Figure S2. I could not review an updated version of figure S2 (not provided). Please, make sure to name GeneX correctly. The scientific community has the right to know what siRNA was used in that experiment. I agree with the authors: the poor quality  of the western blot likely depends on a bad WB antibody.
  3. About Minor point 1. The revised sentence is still wrong. Smoking and occupational exposures don't account for the whole incidence difference between sexes. I would suggest to rephrase as follows: <<The main cause of BC are occupational exposure to urothelial-related carcinogens and smoking. Also, men are more frequently affected than women, with an approximate ratio of 3:1.>>.
  4. About Minor point 3. I could not review this minor point since supplementary files were not made available to me.
  5. About minor point 4 (just a comment). I was mainly concerned about comparing ureters vs tumors. I think authors did a good job comparing tumor-vs-tumor, this comparison was sound.

Author Response

Response to Reviewer 2 Comments

Dear reviewer,

Thank you again for your helpful criticism and the second revision of our manuscript. We accept all points of criticism and have revised our manuscript as follows.

  1. My major point is about the interpretation of the data about the C/T 3'UTR mutation in BC. The luciferase assay was an indirect way of assessing protein levels using a reporter. A transfection- and reporter-based system can only SUGGEST but DOES NOT IMPLY an effect on the expression levels of the protein of interest. Therefore, from my point of view authors are over-interpreting their findings. Moreover, the admission that the C/T 3'UTR mutation-carrying tumor did not show an increased staining of SLC35F2 compared to other NMIBC tumors suggests that this 3'UTR mutation may play a minor role in the biology of SLC35F2. Unfortunately, authors did not consider running further experiments to better address this point.

Thanks for the criticism. You are right, a reporter-based assay can only hypothesize that it has effects on gene and protein expression. The real effect of this 3’UTR mutation depends on many factors that influence or bind the 3’UTR, such as miRNA or mRNA-stabilizing proteins. We did not want to over-interpret the results. Therefore, we write “suggests” instead of “indicates” (line 360) and “hypothesis” instead of “indications” (line 363) in the revised manuscript. We are now also discussing the immunohistochemical staining of SLC35F2 in the mutated tumor in line 365: “Nevertheless, we did not detect an increased average staining in the tumor with the 3’UTR mutation compared to other NMIBC, which suggests that this 3'UTR mutation may play a minor role in the biology of SLC35F2.”

It is not entirely true that we are not considering further experiments. We discussed in the manuscript that the insertion of these mutations in various BC cells by CRISPR/cas is necessary. In addition, we stated in the previous response letter that we are planning to analyze in more detail the effects of mutations in regulation of SLC35F2 and in BC progression. These are complicated and time-consuming experiments and thus we think we cannot include in this manuscript.

  1. About Figure S2. I could not review an updated version of figure S2 (not provided). Please, make sure to name GeneX correctly. The scientific community has the right to know what siRNA was used in that experiment. I agree with the authors: the poor quality of the western blot likely depends on a bad WB antibody.

We are not sure whether the revised supplementary files were correctly submitted with the first revised manuscript. We have resubmitted the files. We have now added the gene name BCL9L instead of GeneX.

  1. About Minor point 1. The revised sentence is still wrong. Smoking and occupational exposures don't account for the whole incidence difference between sexes. I would suggest to rephrase as follows: <<The main cause of BC are occupational exposure to urothelial-related carcinogens and smoking. Also, men are more frequently affected than women, with an approximate ratio of 3:1.>>.

Thank you for your suggestion. We have rephrased it in the revised manuscript.

  1. About Minor point 3. I could not review this minor point since supplementary files were not made available to me.

We are not sure whether the revised supplementary files were correctly submitted with the first revised manuscript. We have resubmitted the files.

  1. About minor point 4 (just a comment). I was mainly concerned about comparing ureters vs tumors. I think authors did a good job comparing tumor-vs-tumor, this comparison was sound.

Thank you for this positive feedback. Nevertheless, the number of analyzed ureters is too small to compare ureters vs. tumor.